# KBioXLM: A Knowledge-anchored Biomedical Multilingual Pretrained Language Model

**Lei Geng**[1]    **Xu Yan**[1]    **Ziqiang Cao**[1*]    **Juntao Li**[1]    **Wenjie Li**[2]    **Sujian Li**[3]
**Xinjie Zhou**[4]    **Yang Yang**[4]    **Jun Zhang**[5]
[1]Institute of Artificial Intelligence, Soochow University    [3]Peking University
[2]The Hong Kong Polytechnic University    [4]Pharmcube    [5]Changping Laboratory
{20215227001, xyannlp}@stu.suda.edu.cn, {zqcao, ljt}@suda.edu.cn
cswjli@comp.polyu.edu.hk, {lisujian, justinzhang}@pku.edu.cn
{zhouxinjie, yangyang}@pharmcube.com

## Abstract

Most biomedical pretrained language models are monolingual and cannot handle the growing cross-lingual requirements. The scarcity of non-English domain corpora, not to mention parallel data, poses a significant hurdle in training multilingual biomedical models. Since knowledge forms the core of domain-specific corpora and can be translated into various languages accurately, we propose a model called KBioXLM, which transforms the multilingual pretrained model XLM-R into the biomedical domain using a knowledge-anchored approach. We achieve a biomedical multilingual corpus by incorporating three granularity knowledge alignments (entity, fact, and passage levels) into monolingual corpora. Then we design three corresponding training tasks (entity masking, relation masking, and passage relation prediction) and continue training on top of the XLM-R model to enhance its domain cross-lingual ability. To validate the effectiveness of our model, we translate the English benchmarks of multiple tasks into Chinese. Experimental results demonstrate that our model significantly outperforms monolingual and multilingual pretrained models in cross-lingual zero-shot and few-shot scenarios, achieving improvements of up to 10+ points. Our code is publicly available at https://github.com/ngwlh-gl/KBioXLM.

## 1 Introduction

In recent years, biomedical pretrained language models (PLMs) (Lee et al., 2020; Gu et al., 2020; Yasunaga et al., 2022) have made remarkable progress in various natural language processing (NLP) tasks. While the biomedical annotation data (Li et al., 2016; Doğan et al., 2014; Du et al., 2019; Collier and Kim, 2004; Gurulingappa et al., 2012) are predominantly in English. Therefore, non-English biomedical natural language process-

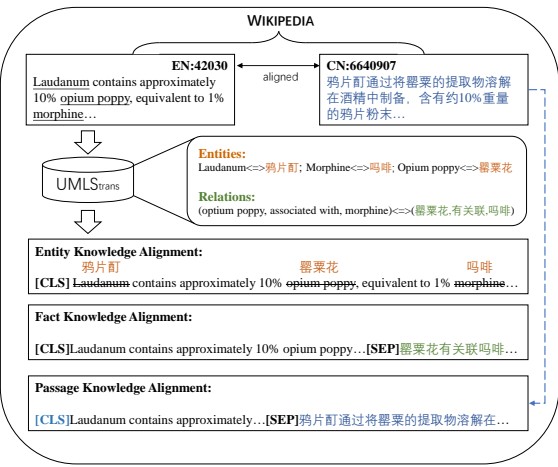

Figure 1: The construction process of aligned biomedical data relies on three types of granular knowledge. We begin with an English passage identified as 42030 , which serves as the primary source. From this passage, we extract entities and relations using UMLS$_{trans}$, and we search for the corresponding aligned Chinese passage 6640907 from Wikipedia. Combining these three granular knowledge sources, we construct aligned corpora. We will predict the relationship between the two passages using the "[CLS]" token.

ing tasks highlight the pressing need for cross-lingual capability. However, most biomedical PLMs focus on monolingual and cannot address cross-lingual requirements, while the performance of existing multilingual models in general domain fall far behind expectations (Devlin et al., 2018; Conneau et al., 2019; Chi et al., 2021). Multilingual biomedical models can effectively tackle cross-lingual tasks and monolingual tasks. Therefore, the development of a multilingual biomedical pretrained model is urgently needed[1].

Unlike in general domains, there is a scarcity of non-English biomedical corpora and even fewer parallel corpora in the biomedical domain, which presents a significant challenge for training mul-

---

*Corresponding author.

[1]Due to the lack of expertise in other languages, we have only conducted experiments in Chinese and English.

tilingual biomedical models. In general domains, back translation (BT) (Sennrich et al., 2015) is commonly used for data augmentation. However, our experiments (refer to "XLM-R+BT" listed in Table 5) reveal that due to the quality issues of domain translation, back translation does not significantly improve multilingual biomedical models' cross-lingual understanding ability. Unlike the entire text, translating entities and relations constituting domain knowledge is unique. Since domain knowledge is considered the most crucial content (Michalopoulos et al., 2020; He et al., 2020), we propose a novel model called KBioXLM to bridge multilingual PLMs like XLM-R (Conneau et al., 2019) into the biomedical domain by leveraging a knowledge-anchored approach. Concretely, we incorporate three levels of granularity in knowledge alignments: entity, fact, and passage levels, to create a text-aligned biomedical multilingual corpus. We first translate the UMLS knowledge base[2] into Chinese to obtain bilingual aligned entities and relations named UMLS$_{trans}$. At the entity level, we employ code-switching (Yang et al., 2020) to replace entities in sentences with expressions in another language according to UMLS$_{trans}$. At the fact level, we transform entity pairs with relationships into another language using UMLS$_{trans}$ and concatenate them after the original monolingual sentence. At the passage level, we collect paired biomedical articles in English and Chinese from Wikipedia[3] to form a coarse-grained aligned corpus. An example of the construction process can be seen in Figure 1. Furthermore, we design three training tasks specifically tailored for the knowledge-aligned data: entity masking, relation masking, and passage relation prediction. It is worth noting that in order to equip the model with preliminary biomedical comprehension ability, we initially pretrain XLM-R on monolingual medical corpora in both Chinese and English. Then, continuously training on top of the model using these three tasks, our approach has the ability to handle cross-lingual biomedical tasks effectively.

To compensate for the lack of cross-lingual evaluation datasets, we translate and proofread four English biomedical datasets into Chinese, involving three different tasks: named entity recognition (NER), relation extraction (RE), and document classification (DC). The experimental results demonstrate that our model consistently outperforms both monolingual and other multilingual pretrained models in cross-lingual zero-shot and few-shot scenarios. On average, our model achieves an impressive improvement of approximately 10 points than general multilingual models. Meanwhile, our method maintains a comparable monolingual ability by comparing common benchmarks in both Chinese and English biomedical domains.

To summarize, our contributions can be outlined as follows:

- We innovatively propose a knowledge-anchored method for the multi-lingual biomedical scenario: achieving text alignment through knowledge alignment.
- We design corresponding tasks for multi-granularity knowledge alignment texts and develop the first multilingual biomedical pre-trained language model to our knowledge.
- We translate and proofread four biomedical datasets to fill the evaluation gap in cross-lingual settings.

## 2 Related Work

### 2.1 Biomedical Pretrained Language Models

In recent years, the advent of pre-trained language models like BERT (Devlin et al., 2018) has brought about a revolution in various downstream NLP tasks, including biomedical research. Researchers such as Lee et al. (2020); Huang et al. (2019); Gu et al. (2020) have focused on training domain-specific PLMs using specialized corpora. For example, PubMedBERT (Gu et al., 2020) constructs a domain-specific vocabulary by leveraging articles from PubMed and is trained on this data. BioLinkBERT (Yasunaga et al., 2022) introduces document-link relation prediction as a pre-training task, achieving state-of-the-art (SOTA) performance on question-answering and document classification tasks. BERT-MK (He et al., 2019) and KeBioLM (Yuan et al., 2021) incorporate UMLS knowledge into the training process. In Chinese, Zhang et al. (2020, 2021b); Cai et al. (2021) have further enhanced BERT with biomedical knowledge augmentation techniques.

While these models have demonstrated impressive results in monolingual settings, many of them lack the ability to handle multilingual tasks effectively. Thus, there is still a need for models that can tackle the challenges presented by multilingual and cross-lingual tasks.

[2]https://www.nlm.nih.gov/research/umls
[3]https://www.wikipedia.org/

| | Tokens |
|---|---|
| **Entity-level** | 31.3M |
| **Fact-level** | 19.6M |
| **Passage-level** | 14M |

Table 1: Tokens of the three-level biomedical multilingual corpus by incorporating three granularity knowledge alignments.

## 2.2 Multi-lingual Pretrained Language Models

Multilingual pre-trained models represent multiple languages in a shared semantic vector space and enable effective cross-lingual processing. Notable examples include mBERT (Devlin et al., 2018), which utilizes Wikipedia data and employs Multilingual Masked Language Modeling (MMLM) during training. XLM (Lample and Conneau, 2019) focuses on learning cross-lingual understanding capability from parallel corpora. ALM (Yang et al., 2020) adopts a code-switching approach for sentences in different languages instead of simple concatenation. XLM-R (Conneau et al., 2019), based on RoBERTa (Liu et al., 2019), significantly expands the training data and covers one hundred languages. To further enhance the cross-lingual transferability of pre-trained models, InfoXLM (Chi et al., 2021) introduces a new pretraining task based on contrastive learning. However, as of now, there is no specialized multilingual model specifically tailored for the biomedical domain.

## 3 Method

This section presents the proposed knowledge-anchored multi-lingual biomedical PLM called KBioXLM. Considering the scarcity of biomedical parallel corpora, we utilize language-agnostic knowledge to facilitate text alignment at three levels of granularity: entity, fact, and passage. Subsequently, we train KBioXLM by incorporating these knowledge-anchored aligned data on the foundation of the multilingual XLM-R model. Figure 2 provides an overview of the training details.

### 3.1 Three Granularities of Knowledge

#### 3.1.1 Knowledge Base

We obtain entity and fact-level knowledge from UMLS and retrieve aligned biomedical articles from Wikipedia.
**UMLS.** UMLS (Unified Medical Language Sys-

tems) is widely acknowledged as the largest and most comprehensive knowledge base in the biomedical domain, encompassing a vast collection of biomedical entities and their intricate relationships. While UMLS offers a broader range of entity types in English and adheres to rigorous classification criteria, it lacks annotated data in Chinese. Recognizing that knowledge often possesses distinct descriptions across different languages, we undertake a meticulous manual translation process. A total of 982 relationship types are manually translated, and we leverage both Google Translator[4] and ChatGPT[5] to convert 880k English entities into their corresponding Chinese counterparts. Manual verification is conducted on divergent translation results to ensure accuracy and consistency. This Chinese-translated version of UMLS is called UMLS$_{trans}$, providing seamless cross-lingual access to biomedical knowledge.

**Wikipedia.** Wikipedia contains vast knowledge across various disciplines and is available in multiple languages. The website offers complete data downloads[6], providing detailed information about each page, category membership links, and interlanguage links. Initially, we collect relevant items in medicine, pharmaceuticals, and biology by following the category membership links. We carefully filter out irrelevant Chinese articles to focus on the desired content by using a downstream biomedical NER model trained on CMeEE-V2 (Zan et al., 2021) dataset. The dataset includes nine major categories of medical entities, including 504 common pediatric diseases, 7085 body parts, 12907 clinical manifestations, and 4354 medical procedures. Then, using the interlanguage links, we associate the textual contents of the corresponding Chinese and English pages, generating aligned biomedical bilingual text.

#### 3.1.2 Entity-level Knowledge

Entities play a crucial role in understanding textual information and are instrumental in semantic understanding and relation prediction tasks. Drawing inspiration from ALM (Yang et al., 2020), we adopt entity-level code-switching and devise the entity masking pretraining objective. The process of constructing entity-level pseudo-bilingual corpora is illustrated in orange in Figure 1. We extract entities from UMLS$_{trans}$ that appear in monolingual

---

[4] https://translate.google.com/
[5] https://openai.com/blog/chatgpt
[6] https://dumps.wikimedia.org/

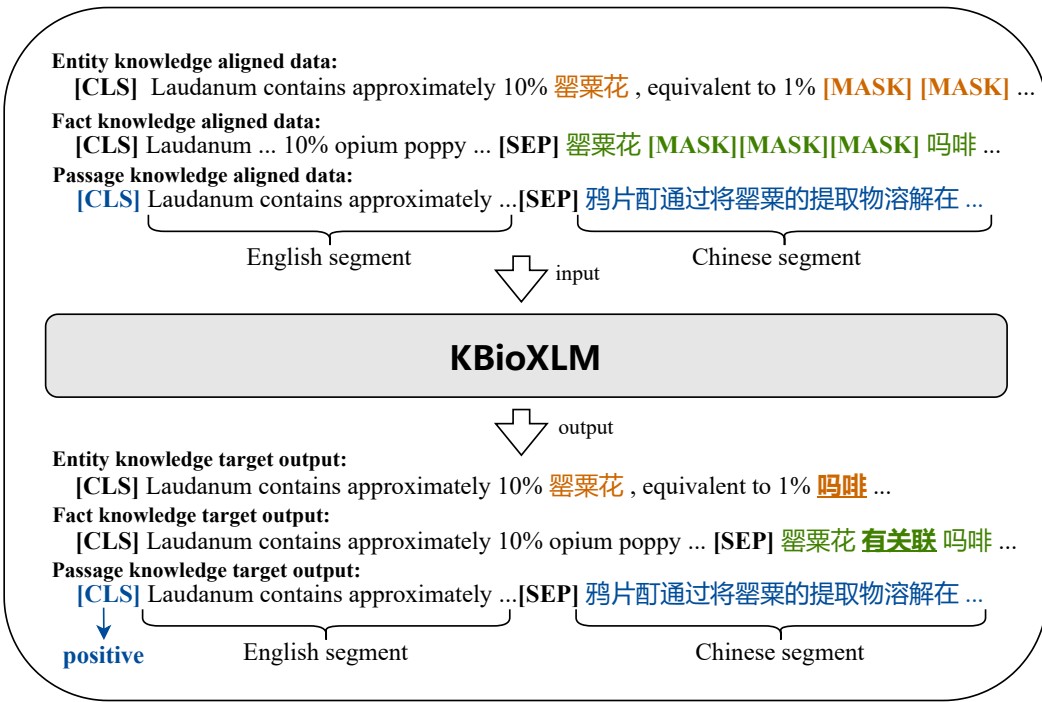

Figure 2: Overview of KBioXLM's training details. Masked entity prediction, masked relation token prediction, and contextual relation prediction tasks are shown in this figure. The color orange represents entity-level pretraining task, green represents fact-level, and blue represents passage-level. Given that the two passages are aligned, our model predicts a "positive" relationship between them.

sentences and their counterparts in the other language. To ensure balance, we randomly substitute 10 biomedical entities with their respective counterparts in each sample, keeping an equal number of replaced entities in both languages.

We design specific pretraining tasks to facilitate the exchange of entity-level information between Chinese and English. Given a sentence $X = \{x_1, x_2, \cdots, x_n\}$ containing Chinese and English entities, we randomly mask 15% of the tokens representing entities in the sentence. The objective of KBioXLM's task is to reconstruct these masked entities. The loss function is defined as follows:

$$L_e = - \sum_i \log P(e_i|X), \tag{1}$$

Here, $e_i$ represents the masked Chinese or English entity.

### 3.1.3 Fact-level Knowledge

Fact refers to a relationship between a subject, predicate, and object in the knowledge base. Our assumption is that if both entities mentioned in the fact are present together in a text, then the text should contain this fact. We employ fact matching to create bilingual corpora and develop a pretraining task called relation masking. The process of

constructing bilingual corpora at the fact level involves the following steps:

- Retrieve potential relationships between paired entities from UMLS$_{trans}$ in monolingual corpus.
- Organize these facts in another language and concatenate them with the original monolingual sentence.

An example is depicted in green color in Figure 1. Given the input text "Laudanum contains approximately 10% opium poppy ...", we extract the fact "(opium poppy, associated with, morphine)" and its corresponding Chinese translation "(罂粟花, 有关联, 吗啡)". The final text would be "Laudanum contains approximately 10% opium poppy ... [SEP] 罂粟花有关联吗啡". We will mask the relationship "有关联".

The fact-level task is to reconstruct the masked relationships. The loss function for relation masking is defined as follows:

$$L_f = - \sum_i \log P(f_i|X), \tag{2}$$

where $f_i$ represents the masked relationship, which can be either a Chinese or an English representation.

### 3.1.4 Passage-level Knowledge

Some biomedical NLP tasks are performed at the document level, so we broaden the scope of cross-lingual knowledge to encompass the passage level. This expansion is illustrated in blue in Figure 1. Specifically, we employ paired biomedical English and Chinese Wikipedia articles to create an aligned corpus at the passage level. This corpus serves as the foundation for designing a pretraining task focused on predicting passage relationships. Inspired by Yasunaga et al. (2022), the strategies employed to construct the passage-level corpus are as follows:

- Randomly selecting one Chinese segment and one English segment, we label them as "positive" if they belong to paired articles and as "random" otherwise.
- We pair consecutive segments in the same language to create contextualized data pairs and label them as "context".

Ultimately, we gather a collection of 30k segment pairs, with approximately equal quantities for each of the three types of segment pairs. The pretraining task employed to incorporate bilingual passage knowledge into the model is passage relationship prediction. The loss function for this task is as follows:

$$L_p = -\log P(c|X_{pair}), \qquad (3)$$

where $c \in \{positive, random, context\}$, $X_{pair}$ is the hidden state with global contextual information.

The tokens present in the three-level biomedical multilingual corpus are documented in Table 1. KBioXLM is trained using an equal proportion of monolingual data and the previously constructed three-level bilingual corpora to ensure the model's proficiency in monolingual understanding. The overall pretraining loss function for KBioXLM is defined as follows:

$$L = L_e + L_f + L_p, \qquad (4)$$

By integrating these three multi-task learning objectives, KBioXLM exhibits improved cross-lingual understanding capability.

### 3.2 Backbone Multilingual Model

Our flexible approach can be applied to any multilingual pre-trained language model. In this study, we adopt XLM-R as our foundational framework, leveraging its strong cross-lingual understanding

capability across various downstream tasks. To tailor XLM-R to the biomedical domain, we conduct additional pretraining using a substantial amount of biomedical monolingual data from CNKI[7] (2.15 billion tokens) and PubMed[8] (2.92 billion tokens). The pre-training strategy includes whole word masking (Cui et al., 2021) and biomedical entity masking. We match the biomedical Chinese entities and English entities contained in UMLS$_{trans}$ with the monolingual corpora in both Chinese and English for the second pretraining task. Clearly, we have already incorporated entity-level knowledge at this pretraining stage to enhance performance. For specific details regarding the pretraining process, please refer to Section A.1. For convenience, we refer to this model as XLM-R+Pretraining.

## 4 Biomedical Dataset Construction

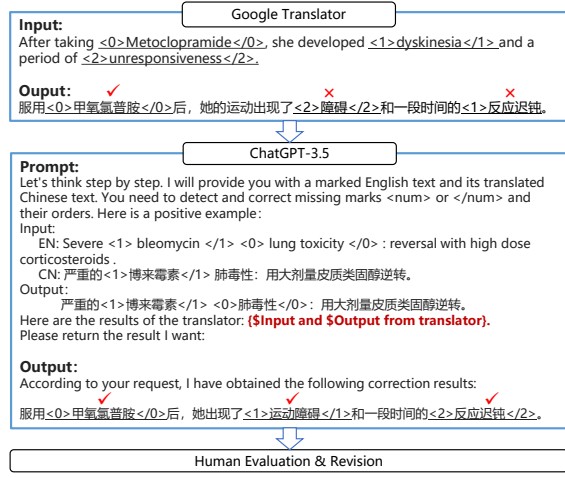

Figure 3: This picture illustrates the sentence, "After taking Metoclopramide, she developed dyskinesia and a period of unresponsiveness." which is initially marked for translation and subsequently revised through a collaborative effort involving ChatGPT and manual editing.

Due to the lack of biomedical cross-lingual understanding benchmarks, we translate several well-known biomedical datasets from English into Chinese by combining translation tools, ChatGPT, and human intervention. As shown in Table 2, these datasets are BC5CDR (Li et al., 2016) and ADE (Gurulingappa et al., 2012) for NER, GAD for RE, and HoC (Hanahan and Weinberg, 2000) for DC. ADE and HoC belong to the BLURB benchmark[9]. Please refer to Section A.2 for details about

---

[7] https://kns.cnki.net/kns8
[8] https://huggingface.co/datasets/pubmed
[9] https://microsoft.github.io/BLURB/

| Dataset | Task | Train | Dev | Test |
|---------|------|-------|-----|------|
| BC5CDR | NER | 500 | 500 | 500 |
| ADE | NER | 3845 | - | 427 |
| GAD† | RE | 4261 | 535 | 534 |
| HoC† | DC | 1295 | 186 | 371 |

Table 2: Statistics of four biomedical cross-lingual datasets. 1.Task types; 2. Number of samples in train/dev/test dataset. 3. The dataset with special mark † belongs to BLURB benchmark.

these four tasks.

The process, as depicted in Figure 3 can be summarized as follows: we conduct a simple translation using Google Translator for the document classification dataset. To preserve the alignment of English entities and their relationships in the NER and RE datasets after translation, we modify them by replacing them with markers "<num>Entity</num>" based on the NER golden labels in the original sentences. Here, "num" indicates the entity's order in the original sentence. During post-processing, we match the translated entities and their relationships back to their original counterparts using the numerical information in the markers. Despite the proficiency of Google Translator, it has some limitations, such as missing entity words, incomplete translation of English text, and semantic gaps. To address these concerns, we design a prompt to leverage ChatGPT in identifying inconsistencies in meaning or incomplete translation between the original sentences and their translations. Subsequently, professional annotators manually proofread the sentences with identified defects while a sample of error-free sentences is randomly checked. This rigorous process guarantees accurate and consistent translation results, ensuring proper alignment of entities and relationships. Ultimately, we obtain high-quality Chinese biomedical datasets with accurately aligned entities and relationships through meticulous data processing.

# 5 Experiments

## 5.1 Dataset and Settings

### 5.1.1 Pretraining

The pretraining process of KBioXLM employs a learning rate of 5e-5, a batch size of 128, and a total of 50,000 training steps. 5,000 warm-up steps are applied at the beginning of the training. The model is pretrained on 4 NVIDIA RTX A5000 GPUs for 14 hours.

| | Language | Biomedical |
|---|----------|------------|
| eHealth | CN | ✓ |
| SMedBERT | CN | ✓ |
| BioBERT | EN | ✓ |
| PubMedBERT | EN | ✓ |
| BioLinkBERT | EN | ✓ |
| mBERT | Multi | ✗ |
| InfoXLM | Multi | ✗ |
| XLM-R | Multi | ✗ |
| XLM-R+BT | Multi | ✓ |
| XLM-R+three KL | Multi | ✓ |
| KBioXLM | Multi | ✓ |

Table 3: Characteristics of our baselines. "✓" indicates that the model has this feature while "✗" means the opposite. "CN" means Chinese, "EN" represents English and "Multi" represents Multilingual.

### 5.1.2 Finetuning

For monolingual tasks and cross-lingual understanding downstream tasks listed in Table 2, the backbone of Named Entity Recognition is the encoder part of the language model plus conditional random fields (Lafferty et al., 2001). The simplest sequence classification is employed for relation extraction and document classification tasks. F1 is used as the evaluation indicator for these tasks.

## 5.2 Baselines

To compare the performance of our model in monolingual comprehension tasks, we select SOTA models in English and Chinese biomedical domains. Similarly, to assess our model's cross-lingual comprehension ability, we conduct comparative experiments with models that possess strong cross-lingual understanding capability in general domains, as there is currently a lack of multilingual PLMs specifically tailored for the biomedical domain.

**Monolingual Biomedical PLMs.** For English PLMs, we select BioBERT (Lee et al., 2020), PubMedBERT (Gu et al., 2020), and BioLinkBERT (Yasunaga et al., 2022) for comparison, while for Chinese, we choose eHealth (Wang et al., 2021) and SMedBERT (Zhang et al., 2021b).

**Multilingual PLMs.** XLM-R baseline model and other SOTA multilingual PLMs, including mBERT (Devlin et al., 2018), InfoXLM (Chi et al., 2021) are used as our baselines. We also compare the results of two large language models (LLMs) that currently perform well in generation tasks on these four tasks, namely ChatGPT and ChatGLM-6B[10].

---

[10] https://chatglm.cn/blog

| | Datasets | LLMs | | Multilingual PLMs | | | Multilingual Biomedical PLMs | | |
|---|---|---|---|---|---|---|---|---|---|
| | | ChatGPT | ChatGLM | mBERT | InfoXLM | XLM-R | XLM-R+BT | XLM-R+three KL | KBioXLM |
| EN-to-CN | ADE | 64.50 | 27.30 | 59.43 | 64.23 | 57.62 | 64.78 | 65.47 | **70.88** |
| | BC5CDR | 63.00 | 40.70 | 54.59 | 66.39 | 57.43 | 63.83 | 70.64 | **73.02** |
| | GAD | 48.30 | 48.30 | 59.52 | 67.03 | 68.79 | 70.65 | 75.29 | **78.91** |
| | HoC | 35.10 | 28.00 | 14.29 | 44.83 | 37.58 | 60.45 | 68.09 | **78.83** |
| | AVG | 52.73 | 36.08 | 46.96 | 60.62 | 55.36 | 64.93 | 69.87 | **75.41** |
| CN-to-EN | ADE | 71.79 | 42.90 | 71.31 | 79.52 | 77.38 | 77.92 | 78.07 | **85.61** |
| | BC5CDR | 54.80 | 38.80 | 64.12 | 75.36 | 72.40 | 76.29 | 78.92 | **84.52** |
| | GAD | 51.20 | 52.20 | 64.16 | 69.69 | 69.96 | 74.74 | 76.36 | **81.16** |
| | HoC | 41.30 | 31.70 | 37.08 | 48.34 | 37.67 | 57.65 | 61.09 | **73.99** |
| | AVG | 54.77 | 41.40 | 59.17 | 68.23 | 64.35 | 71.65 | 73.61 | **81.32** |

Table 4: Cross lingual zero shot results. "EN-to-CN" and "CN-to-EN" indicate training in English and testing on Chinese datasets, and vice versa. AVG represents the average F1 score across four cross-lingual tasks.

| | | 10-shot | | | | | 100-shot | | | | |
|---|---|---|---|---|---|---|---|---|---|---|---|
| | | ADE | BC5CDR | GAD | HoC | AVG | ADE | BC5CDR | GAD | HoC | AVG |
| CN Bio PLMs | eHealth | 64.57 | 67.10 | 67.57 | 62.08 | 65.33 | 78.39 | 79.09 | 72.83 | 74.66 | 76.24 |
| | SMedBERT | 54.61 | 61.11 | 67.61 | 33.17 | 54.13 | 75.18 | 75.25 | 69.36 | 66.24 | 71.51 |
| Multi PLMs | mBERT | 57.88 | 55.15 | 66.44 | 44.61 | 56.02 | 69.88 | 74.11 | 73.65 | 67.08 | 71.18 |
| | InfoXLM | 66.92 | 62.47 | 69.48 | 57.47 | 64.09 | 76.35 | 75.62 | 76.19 | 67.70 | 73.97 |
| | XLM-R | 61.02 | 56.66 | 73.85 | 43.74 | 58.82 | 73.06 | 72.93 | 76.18 | 64.12 | 71.57 |
| LLMs | ChatGPT | 66.20 | 60.40 | 49.40 | 50.20 | 56.55 | - | - | - | - | - |
| | ChatGLM | 26.20 | 27.10 | 52.60 | 23.50 | 32.35 | - | - | - | - | - |
| Multi Bio PLMs | XLM-R+three KL | 71.50 | 71.01 | 76.36 | 73.45 | 73.08 | 76.10 | 77.19 | 77.86 | 76.89 | 77.01 |
| | KBioXLM | **75.49** | **74.98** | **80.76** | **79.41** | **77.66** | **79.45** | **80.63** | **81.98** | **83.20** | **81.32** |

Table 5: Cross lingual EN-to-CN few shot results. "Bio" represents Biomedical and "Multi" represents Multilingual. Due to the limited number of input tokens, we only conduct 10-shot experiments for LLMs.

**Multilingual Biomedical PLMs.** To our knowledge, there is currently no multilingual pretraining model in biomedical. Therefore, we build two baseline models on our own. Considering the effectiveness of back-translation (BT) as a data augmentation strategy in general multilingual pretraining, we train baseline XLM-R+BT using back-translated biomedical data. Additionally, in order to assess the impact of additional pretraining in KBioXLM, we directly incorporate the three levels of knowledge mentioned earlier into the XLM-R architecture, forming XLM-R+three KL.

Please refer to Table 3 for basic information about our baselines.

### 5.3 Main Results

This section explores our model's cross-lingual and monolingual understanding ability. Please refer to Appendix A.3 for the model's monolingual understanding performance on more tasks.

#### 5.3.1 Cross-lingual Understanding Ability

We test our model's cross-lingual understanding ability on four tasks in two scenarios: zero-shot

and few-shot. As shown in 4 and 5, KBioXLM achieves SOTA performance in both cases. Our model has a strong cross-lingual understanding ability in the zero-shot scenario under both "EN-to-CN" and "CN-to-EN" settings. It is worth noting that the performance of LLMs in language understanding tasks is not ideal. Moreover, compared to ChatGPT, the generation results of ChatGLM are unstable when the input sequence is long. Similarly, general-domain multilingual PLMs also exhibit a performance difference of over 10 points compared to our model. The poor performance of LLMs and multilingual PLMs underscores the importance of domain adaptation. XLM-R+three KL is pretrained with just 65M tokens, and it already outperforms XLM-R by 14 and 9 points under these two settings. And compared to XLM-R+BT, there is also an improvement of 5 and 2 points, highlighting the importance of knowledge alignment. Compared to KBioXLM, XLM-R+three KL performs 5 or more points lower. This indicates that excluding the pretraining step significantly affects the performance of biomedical cross-lingual understanding tasks,

| | Dataset | ADE | BC5CDR | GAD | HoC | AVG |
|---|---|---|---|---|---|---|
| **EN-to-EN** | **EN Bio PLMs** Pubmedbert | 90.22 | 89.37 | 80.50 | 83.97 | 86.02 |
| | BioBERT | 89.85 | 87.14 | 83.95 | 82.44 | 85.85 |
| | BioLinkBERT | 90.12 | **89.76** | **85.90** | **84.53** | **87.58** |
| | **Multi PLMs** mBERT | 89.61 | 84.44 | 79.67 | 80.04 | 83.44 |
| | InfoXLM | 89.91 | 86.50 | 77.33 | 78.70 | 83.11 |
| | XLM-R | 90.07 | 85.92 | 80.45 | 80.12 | 84.14 |
| | **Multi Bio PLMs** XLM-R+three KL | 89.83 | 86.50 | 82.35 | 82.08 | 85.19 |
| | KBioXLM | **90.56** | 88.87 | 83.04 | 83.66 | 86.53 |
| **CN-to-CN** | **CN Bio PLMs** eHealth | **85.20** | 72.22 | 77.53 | 78.67 | 78.41 |
| | SMedBERT | 84.30 | 41.30 | 74.34 | 79.27 | 69.80 |
| | **Multi PLMs** mBERT | 83.27 | 67.83 | 75.69 | 76.35 | 75.79 |
| | InfoXLM | 83.48 | 72.00 | 71.71 | 78.53 | 76.43 |
| | XLM-R | 82.67 | 70.57 | 78.13 | 78.52 | 77.47 |
| | **Multil Bio PLMs** XLM-R+three KL | 83.26 | 78.09 | 79.61 | 80.73 | 80.42 |
| | KBioXLM | 83.02 | **78.94** | **82.76** | **83.47** | **82.05** |

Table 6: The performance of KBioXLM and our baselines on English and Chinese monolingual comprehension tasks.

| Datasets | ADE | BC5CDR | GAD | HoC |
|---|---|---|---|---|
| **KBioXLM** | **70.88** | **73.02** | **78.91** | **78.83** |
| **w/o Pas** | 69.99 | 71.55 | 78.08 | 77.28 |
| **w/o Pas+Fact** | 70.20 | 69.78 | 76.87 | 76.92 |
| **w/o Pas+Fact+Ent** | 66.61 | 67.03 | 75.93 | 76.74 |

Table 7: KBioXLM's cross-lingual understanding ablation experiments in the zero-shot scenario.

highlighting the importance of initially pretraining XLM-R to enhance its biomedical understanding capability.

In the "EN-to-CN" few-shot scenario, we test models' cross-lingual understanding ability under two settings: 10 training samples and 100 training samples. It also can be observed that XLM-R+three KL and KBioXLM perform the best among these four types of PLMs. Multilingual PLMs and Chinese biomedical PLMs have similar performance. However, compared to our method, there is a difference of over 10 points in the 10-shot scenario and over 5 points in the 100-shot scenario. This indicates the importance of both domain-specific knowledge and multilingual capability, which our model satisfies.

### 5.3.2 Monolingual Understanding Ability

Although the focus of our model is on cross-lingual scenarios, we also test its monolingual comprehension ability on these four datasets. Table 6 shows the specific experimental results. It can be seen that KBioXLM can defeat most other PLMs in these tasks, especially in the "CN-to-CN" scenario. Compared to XLM-R, KBioXLM has an average

improvement of up to 4 points. BioLinkBERT performs slightly better than ours on English comprehension tasks because it incorporates more knowledge from Wikipedia. KBioXLM's focus, however, lies in cross-lingual scenarios, and we only utilize a small amount of aligned Wikipedia articles.

### 5.4 Ablation Study

This section verifies the effectiveness of different parts of the used datasets. Table 7 presents the results of ablation experiments in zero-shot scenarios. Removing the bilingual aligned data at the passage level results in a 1-point decrease in model performance across all four tasks. Further removing the fact-level data leads to a continued decline. When all granularity bilingual knowledge data is removed, our model's performance drops by approximately 4 points. These experiments demonstrate the effectiveness of constructing aligned corpora with three different granularities of knowledge. Due to the utilization of entity knowledge in the underlying XLM-R+pretraining, it is difficult to accurately assess the performance when none of the three types of knowledge are used.

## 6 Conclusion

This paper proposes KBioXLM, a model that transforms the general multi-lingual PLM into the biomedical domain using a knowledge-anchored approach. We first obtain biomedical multilingual corpora by incorporating three levels of knowledge alignment (entity, fact, and passage) into the monolingual corpus. Then we design three train-

ing tasks, namely entity masking, relation masking, and passage relation prediction, to enhance the model's cross-lingual ability. KBioXLM achieves SOTA performance in cross-lingual zero-shot and few-shot scenarios. In the future, we will explore biomedical PLMs in more languages and also venture into multilingual PLMs for other domains.

## Limitations

Due to the lack of proficient personnel in other less widely spoken languages, our experiments were limited to Chinese and English only. However, Our method can be applied to various other languages, which is highly significant for addressing cross-lingual understanding tasks in less-resourced languages. Due to device and time limitations, we did not explore our method on models with larger parameter sizes or investigate cross-lingual learning performance on generative models. These aspects are worth exploring in future research endeavors.

## Acknowledgements

We thank all reviewers for their valuable comments. This work was supported by the Young Scientists Fund of the National Natural Science Foundation of China (No. 62106165).

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

# A    Appendix

## A.1    XLM-R Pretraining Settings

Building upon XLM-R, we train XLM-R using medical data from CNKI of 2.15B tokens and data from PubMed of 2.92B tokens. During the training process, we initialize the model parameters with XLM-R. It is important to note that in order to speed up the training process, we first calculate the distribution of tokens from cnki and pubmed in the XLM-R vocabulary. Then, we utilize a one-hot matrix to reduce the original MLM head of XLM-R from $250002 \times 768$ to $37030 \times 768$, and use it as the new MLM head. The pre-training strategy includes whole word masking (Cui et al., 2021) and biomedical entity masking. We match the biomedical Chinese entities and English entities contained in $UMLS_{trans}$ with the monolingual corpora in both Chinese and English for the second pretraining task. The proportion of masked tokens in the sentence is the same as XLM-R, and both strategies masked tokens at a 1:1 ratio. We limit the masked biomedical entities to a maximum length of 3 to accelerate the model's learning process. The peak learning rate for this training process is set to 1e-4, with a batch size of 1280 and a total of 150,000 training steps. In the first 10,000 steps, the learning rate linearly increases. The model was pretrained on 4 NVIDIA RTX A5000 GPUs for two weeks.

## A.2    Four Downstream Tasks

**BC5CDR.** BC5CDR comprises a collection of 1500 PubMed abstracts[11] and has been preprocessed by Christopoulou et al. (2019). The objective of the model is to identify two distinct entity types in the text: chemical and disease.

**ADE.** ADE is another NER dataset sourced from PubMed documents, primarily focused on identifying drugs and adverse effects entities. We leverage the dataset provided in SpERT (Eberts and Ulges, 2019).

**GAD.** GAD serves the purpose of detecting the association between gene entities and disease entities in a given sentence. The gene and disease entities within the sentences are denoted by special

[11] https://pubmed.ncbi.nlm.nih.gov/

| Dataset | Task | Train | Dev | Test |
|---------|------|-------|-----|------|
| CMeEE (Zan et al., 2021) | NER | 15000 | 5000 | 3000 |
| CMeIE(Guan et al., 2020) | IE | 14339 | 3585 | 4482 |
| CHIP-CDN | Diagnosis Normalization | 6000 | 2000 | 10192 |
| CHIP-STS | Sentence Similarity | 16000 | 4000 | 10000 |
| CHIP-CTC (Zong et al., 2021) | Sentence Classification | 22962 | 7682 | 10000 |
| KUAKE-QIC | Intent Classification | 6931 | 1955 | 1994 |
| KUAKE-QTR | Query-Document Relevance | 24174 | 2913 | 5465 |
| KUAKE-QQR | Query-Query Relevance | 15000 | 1600 | 1596 |

Table 8: Statistics of Chinese biomedical datasets. 1.Task types; 2. Number of samples in train/dev/test dataset.

| Dataset | Task | Train | Dev | Test |
|---------|------|-------|-----|------|
| BC5-chem† (Li et al., 2016) | | 4560 | 4581 | 4797 |
| BC5-disease† (Li et al., 2016) | | 4560 | 4581 | 4797 |
| NCBI-disease† (Doğan et al., 2014) | | 5424 | 923 | 940 |
| BC2GM† (Du et al., 2019) | | 12500 | 2500 | 5000 |
| JNLPBA† (Collier and Kim, 2004) | NER | 16807 | 1739 | 3856 |
| BC5CDR (Li et al., 2016) | | 500 | 500 | 500 |
| ADE (Gurulingappa et al., 2012) | | 3845 | - | 427 |
| CHR (Sahu et al., 2019) | | 7298 | 1182 | 3614 |
| BioRED (Luo et al., 2022) | | 400 | 100 | 100 |
| ChemProt† (Krallinger et al., 2017) | | 18035 | 11268 | 15745 |
| DDI† (Herrero-Zazo et al., 2013) | | 25296 | 2496 | 5716 |
| GAD† (Bravo et al., 2015) | RE | 4261 | 535 | 534 |
| AIMed (Bunescu et al., 2005) | | 5251 | - | 583 |
| HoC† (Hanahan and Weinberg, 2000) | DC | 1295 | 186 | 371 |

Table 9: Statistics of English biomedical datasets. 1.Task types; 2. Number of samples in train/dev/test dataset; 3. The dataset with special mark † belongs to BLURB benchmark.

| Dataset | ehealth | SMedBERT | XLM-R | KBioXLM |
|---------|---------|----------|-------|---------|
| CMeEE-V2 | 59.56 | **59.86** | 57.94 | 59.48 |
| CMeIE-V2 | 49.74 | 48.72 | **50.32** | 50.22 |
| CHIP-CDN | **59.32** | 55.46 | 51.04 | 57.89 |
| CHIP-STS | **85.48** | 84.98 | 81.71 | 83.20 |
| CHIP-CTC | 63.15 | **68.05** | 58.39 | 66.30 |
| KUAKE-QIC | 85.66 | **85.71** | 83.10 | 82.85 |
| KUAKE-QTR | **64.56** | 61.59 | 58.23 | 59.49 |
| KUAKE-QQR | **85.65** | 82.83 | 80.52 | 81.89 |
| AVG | **69.14** | 68.40 | 65.16 | 67.67 |

Table 10: Monolingual Chinese results.

markers, namely "@GENE$" and "@DISEASE$", respectively.

**HoC.** The Hallmarks of Cancer (HoC) corpus comprises PubMed abstracts with binary labels indicating specific cancer hallmarks. It contains 37 detailed hallmarks grouped into ten top-level categories. Models are required to predict these 10 top-level categories.

Please refer to Table 2 for quantity statistics.

## A.3    Monolingual Biomedical Tasks

The Chinese Biomedical Language Understanding Evaluation (CBLUE) (Zhang et al., 2021a) benchmark[12] in the Chinese medical domain includes

[12] https://github.com/CBLUEbenchmark/CBLUE

| Dataset | PubMedBERT | BioLinkBERT | XLM-R | KBioXLM |
|---|---|---|---|---|
| BC5-chem | **93.08** | 92.97 | 88.74 | 91.84 |
| BC5-disease | 85.52 | **85.78** | 81.78 | 84.91 |
| Ncbi-disease | 86.60 | 86.69 | **86.85** | 86.67 |
| BC2GM | 83.74 | **84.33** | 81.91 | 83.05 |
| JNLPBA | 79.16 | 78.89 | 79.32 | **79.42** |
| BC5CDR | 89.37 | **89.76** | 85.92 | 88.87 |
| ADE | 90.22 | 90.12 | 90.07 | **90.56** |
| CHR | 91.61 | 91.16 | 91.08 | **91.82** |
| BioRED | 90.74 | **91.14** | 84.54 | 88.72 |
| ChemProt | **77.95** | 76.76 | 68.49 | 76.68 |
| DDI | **81.02** | 79.57 | 74.30 | 79.43 |
| GAD | 80.50 | **85.90** | 80.45 | 83.04 |
| AIMed | **88.41** | 85.96 | 70.37 | 84.26 |
| HoC | 83.97 | **84.53** | 80.12 | 83.66 |
| AVG | 85.85 | **85.97** | 81.46 | 85.27 |

Table 11: Monolingual English results.

tasks such as NER, RE, sentence classification, and more. Similarly, the English medical domain also encompasses these tasks. The quantity statistics for the Chinese and English datasets are shown in Table 8 and Table 9, respectively.

Here, we compare KBioXLM with two SOTA Chinese biomedical models, eHealth and SMed-BERT on CBLUE benchmark and the English SOTA models, PubMedBERT and BioLinkBERT on the corresponding English biomedical tasks.

Table 10 and Table 11 represent the results of evaluating the model's monolingual comprehension ability on Chinese and English biomedical benchmarks, respectively. Our model KBioXLM shows significant improvements of two points and four points compared to XLM-R on average, respectively. This indicates that compared to general multilingual PLMs, our model has stronger biomedical comprehension capability. However, as our model primarily focuses on addressing cross-lingual understanding tasks, it falls slightly behind the current SOTA monolingual biomedical models.