# OpenReview forum: "KBioXLM: A Knowledge-anchored Biomedical Multilingual Pretrained Language Model"
_EMNLP/2023/Conference — EMNLP 2023 Findings_

### Official Review · Reviewer_eLNp · 2023-08-04

**Soundness:** 3

**Excitement:**

4: Strong: This paper deepens the understanding of some phenomenon or lowers the barriers to an existing research direction.

**Paper Topic And Main Contributions:**


The authors present the KBioXLM, a bilingual model (English and Chinese) for the biomedical domain. It is based on the multilingual XLM-R model and incorporates three levels of knowledge (entity, fact, passage).

**Questions For The Authors:**


1. The model is currently only for English and Chinese, so it is rather bilingual than multilingual.

2. The introduction is confusing, there is a lot about methods and results, and not much about why building a biliangual model, instead of relying on a cross-ligual, and advantages/disadvantages of each of them.

3. Methods: I miss a schema/figure of the approach and more details about the manual verification for divergences (Page 3, lines 203-205). Further, in section 3.2, I wonder why th authors did no rely on bilingual documents from PubMed, but just the monolinhual ones.



**Reasons To Accept:**


- A new bilingual model for en/zh that outperformed previous works for various NLP tasks and datasets.

**Reasons To Reject:**


- They refer to bilingual model, but it is rather only multilingual.
- I miss more motivation on building the bilingual model instead of relying on a cross-lingual model

**Reproducibility:**

2: Would be hard pressed to reproduce the results. The contribution depends on data that are simply not available outside the author's institution or consortium; not enough details are provided.

**Reviewer Confidence:**

2: Willing to defend my evaluation, but it is fairly likely that I missed some details, didn't understand some central points, or can't be sure about the novelty of the work.

**Typos Grammar Style And Presentation Improvements:**


- Page 1, line 34: "While, " -> remove comma

- Page 1, lines 43-45: Maybe put the references at the end of the sentence, not in the middle.

- Page 1, lines 54-59: this passage describes results in the motivation section, it's better left for later.

- Page 2, line 65: Since the proposed system is based on XLM-R, it coud have been introduced with some details before.

- Page 3, line 200: does expertly translated means manually?

- Page 3, lines 219-220: please add more details about the dataset.

- Table 3: throughout the paper, please use either "cn" or "zh" for Chinese.

---

> ### Author Rebuttal · Authors · 2023-08-28
>
> We greatly appreciate your valuable and comprehensive feedback on our paper.  Your thoughtful observations and suggestions have played a pivotal role in shaping the direction of our research. We value your expertise and commitment to helping us improve our work.
>
> 1."The model is currently only for English and Chinese, so it is rather bilingual than multilingual."
>
> I want to express my gratitude for your notice of this issue.
> Our base model XML-R is multilingual, giving our model the potential to support multiple languages. Additionally, the approach we've employed for incorporating knowledge also caters to multiple languages. However, as stated in the "Limitations" section, due to the lack of expertise in other languages, we have only conducted experiments in Chinese and English. We will acknowledge this limitation using a footnote in the introduction. In the future, we plan to explore the cross-lingual capabilities of testing our model in other language datasets that align with English dataset.
>
> 2."The introduction is confusing, there is a lot about methods and results, and not much about why building a biliangual model, instead of relying on a cross-ligual , and advantages/disadvantages of each of them."
>
> Your mention of this problem is valued and acknowledged.
> Firstly, bilingual models can effectively tackle cross-lingual tasks. Furthermore, Chinese-only and English-only tasks are also essential, and we have conducted experiments in the appendix to validate their effectiveness. However, it's evident that cross-lingual capabilities hold more practical value, and we intend to revise the introduction to include additional relevant descriptions.
>
> 3."Methods: I miss a schema/figure of the approach and more details about the manual verification for divergences (Page 3, lines 203-205). Further, in section 3.2, I wonder why th authors did no rely on bilingual documents from PubMed , but just the monolinhual ones."
>
> I'm obliged to you for pointing out this issue.
> Firstly, regarding the issue of manual verification, we invite experts with medical knowledge to assist us in our translation work.
>
> Secondly, thank you very much for providing us with the source of the bilingual PubMed data. Currently, our paper has only utilized parallel Wikipedia data for passage-level knowledge. In the future, we plan to retrieve bilingual PubMed articles to incorporate into our work. The main purpose of section 3.2 is to transition XLM-R from the general field to the medical field. We found that PubMed mainly collects English medical literature and organizes it in a unified data format on huggingface, while CNKI has a more comprehensive collection in Chinese medical literature. Due to our previous data preparation work, we have also accumulated rich CNKI corpus, so we chose CNKI.
>
> 4."Typos Grammar Style And Presentation Improvements"
>
> Thank you very much for providing these suggestions.
> We will address the issues you have pointed out in the next version.

---

### Official Review · Reviewer_yywL · 2023-08-04

**Soundness:** 4

**Excitement:**

3: Ambivalent: It has merits (e.g., it reports state-of-the-art results, the idea is nice), but there are key weaknesses (e.g., it describes incremental work), and it can significantly benefit from another round of revision. However, I won't object to accepting it if my co-reviewers champion it.

**Paper Topic And Main Contributions:**

This work proposes a knowledge-centric pretraining strategy specifically aims to improve data efficient in multilingual pertaining on the biomedical domain. Comprehensive experiments on downstream bilingual benchmarks show certain method does reduce dependency on parallel datasets while obtain good performance on biomedical benchmarks.


**Questions For The Authors:**

See weakness.

**Reasons To Accept:**

1. Providing a valuable insight that knowledge (which is usually multilingual in the KB) is essential for bridging multilingual abilities.
2. Performance of KBioXLM significantly higher than other multilingual biomedical PLMs on biomedical benchmarks.
3. Proposing a multi-level knowledge-enhanced pretraining method.

**Reasons To Reject:**

  1.  As authors use both English and Chinese Wikipedia to build the knowledge-enhanced dataset, it is natural to be curious about how a multilingual pertaining language model directly trained by this bilingual corpus will perform in benchmarks mentioned in the article.
  2.  It’s hard to understand why the fact knowledge task works. Actually, the improvement brought by the Fact Knowledge task seems to be minor compared with Pas & Ent based on Table 7. It would be appreciated if the authors can provide more discussion about this task.
  3.  Adding English-only and Chinese-only biomedical PLM in results along with multilingual PLMs may provide more insights.

**Reproducibility:**

4: Could mostly reproduce the results, but there may be some variation because of sample variance or minor variations in their interpretation of the protocol or method.

**Reviewer Confidence:**

4: Quite sure. I tried to check the important points carefully. It's unlikely, though conceivable, that I missed something that should affect my ratings.

---

> ### Author Rebuttal · Authors · 2023-08-28
>
> We extend our heartfelt appreciation to you for your meticulous evaluation of our manuscript. Your constructive criticisms and valuable insights have immensely aided in refining the quality of our research. We are thankful for your dedication to providing thorough feedback.
>
> 1."As authors use both English and Chinese Wikipedia to build the knowledge-enhanced dataset, it is natural to be curious about how a multilingual pertaining language model directly trained by this bilingual corpus  will perform in benchmarks mentioned in the article."
>
> I'm grateful for you highlighting this concern.
> This dataset only contains 14M tokens, which is insufficient for pretraining a language model.  Taking into account the English biomedical pretrained language model "PubMedBERT", which was trained on a medical corpus containing 3B tokens.
>
> 2."It’s hard to understand why the fact knowledge task works. Actually, the improvement brought by the Fact Knowledge task seems to be minor compared with Pas & Ent based on Table 7. It would be appreciated if the authors can provide more discussion about this task."
>
> It's great that you've flagged up this issue.
> In biomedical texts, entities and their relationships are particularly important for understanding tasks, and the biomedical knowledge base records the rich relationships between entity pairs. Therefore, we came up with the idea of constructing fact granularity aligned corpora and designing relationship token prediction tasks to learn the internal relationships between entities, which is particularly helpful for downstream relation extraction(RE) tasks. Kindly review the performance of the GAD dataset as displayed in the table below. (GAD is a RE task).
>
> |       |  BC5CDR   | GAD  |
> |  ----  | ----  | ---- |
> | KBioXLM | 73.02 | 78.91 |
> | w/o Pas+Fact+Ent  | 67.03 | 75.93 |
> | w only Fact | 70.17 | 77.68 |
>
>
> 3."Adding English-only and Chinese-only biomedical PLM in results along with multilingual PLMs may provide more insights."
>
> I'm thankful for your recognition of this matter.
> In a fully fine-tuning monolingual scenario, we also conducted experimental comparisons on many different types of tasks. Please refer to Tables 10 and 11 in the appendix. Our model maintains excellent monolingual comprehension performance in English compared with English biomedical SOTA models. Our model performs slightly less effectively in Chinese. However, this is partly due to the fact that training data for other Chinese medical pretrained language models is proprietary and closely aligned with the testing tasks, such as medical consultation data.
>
> We learn the model in English and test it in Chinese in a 10-shot scenario. And compared the performance of Chinese only biomedical PLM in this scenario. Please refer to Table 5 for relevant experiments. "CN Bio PLMs" represent Chinese only biomedical PLMs, including eHealth and SMedBERT. It can be observed that our model performs much better than the current Chinese biomedical monolingual SOTA model in the few shot scenario.
>
> In conclusion, in the domain of medicine, the majority of literature is in English, and benchmarks in other languages are scarce. Hence, the model's cross-lingual capability is of greater significance. Therefore, in the main experiments, we have contrasted the model's cross-lingual comprehension capabilities.

---

### Official Review · Reviewer_VPvR · 2023-08-05

**Soundness:** 4

**Excitement:**

4: Strong: This paper deepens the understanding of some phenomenon or lowers the barriers to an existing research direction.

**Paper Topic And Main Contributions:**

The paper introduces KBioXLM, a knowledge-anchored approach to transform the multilingual pretrained model XLM-R for handling biomedical domain tasks across languages. The main contributions are:

- Addressing the scarcity of non-English domain corpora and the absence of parallel data in the biomedical domain.
- Validating the model through translated English benchmarks into Chinese

**Reasons To Accept:**

- Well-motivated paper shedding light on the biomedical applications for minor languages.
- Demonstrates impressive performance improvement (up to 10+ points) in cross-lingual zero-shot and few-shot scenarios.

**Reasons To Reject:**

- Limited evaluation scope by only testing on Chinese, additional languages should be considered for a more comprehensive evaluation.
- Investigating the performance of the model when there are fewer tokens at each granularity level could provide additional insights.


**Reproducibility:**

3: Could reproduce the results with some difficulty. The settings of parameters are underspecified or subjectively determined; the training/evaluation data are not widely available.

**Reviewer Confidence:**

3: Pretty sure, but there's a chance I missed something. Although I have a good feel for this area in general, I did not carefully check the paper's details, e.g., the math, experimental design, or novelty.

**Typos Grammar Style And Presentation Improvements:**

Ablation tests considering situations with only one or two granularity levels (entity, fact, or passage) would further validate the proposed approach. (An extension of Table 7)

---

> ### Author Rebuttal · Authors · 2023-08-28
>
> We would like to express our sincere gratitude to you for your insightful and comprehensive feedback on our manuscript. your thoughtful comments and suggestions have significantly contributed to the enhancement of our work. We are truly appreciative of your time and effort in reviewing our paper.
>
> 1."Limited evaluation scope by only testing on Chinese, additional languages should be considered for a more comprehensive evaluation."
>
> Thank you for pointing out this issue.
> Our base model XML-R is multilingual, giving our model the potential to support multiple languages. Additionally, the approach we've employed for incorporating knowledge also caters to multiple languages. However, as stated in the "Limitations" section, due to the lack of expertise in other languages, we have only conducted experiments in Chinese and English. We will acknowledge this limitation using a footnote in the introduction. In the future, we plan to explore the cross-lingual capabilities of testing our model in other language datasets that align with English dataset.
>
> 2."Investigating the performance of the model when there are fewer tokens at each granularity level could provide additional insights."
>
> I appreciate you bringing this matter to my attention.
> We attempt to reduce the number of tokens at each granularity to 1/5 of the original number of tokens, and once again verify the cross language understanding performance of the model. The results are presented in the table "less tokens" below. When we reduce the number of training tokens to the original 20%, the performance of the model significantly decreases, but it still outperforms the baseline model. From the experiment, it is not difficult to see that the knowledge granularity aligned corpus and its training method proposed in this article are effective, and the training token is positively correlated with the performance of the model.
>
> |       |  BC5CDR   | GAD  |
> | ---- |  ----  | ----  |
> | KBioXLM | 73.02 | 78.91 |
> | w/o Pas+Fact+Ent  | 67.03 | 75.93 |
> | Loss tokens | 70.22 | 76.52 |
>
> 3."Ablation tests considering situations with only one or two granularity levels (entity, fact, or passage) would further validate the proposed approach. (An extension of Table 7)"
>
> Your observation of this matter is duly noted and appreciated.
> We validated the effectiveness of each granularity separately, and the results are presented in the table below. It can be seen that compared to the baseline model, adding any granularity of knowledge alignment corpus can help improve the cross language understanding performance of the model.
>
> |       |  BC5CDR   | GAD  |
> | ---- |  ----  | ----  |
> | KBioXLM | 73.02 | 78.91 |
> | w/o Pas+Fact+Ent  | 67.03 | 75.93 |
> | w only Ent | 69.78 | 76.87 |
> | w only Fact | 70.17 | 77.68 |
> | w only Pas | 69.73 | 77.19 |

---

### Meta-Review · Area_Chair_u98o · 2023-09-20

**Recommendation:** 5

**Metareview:**

Reviewers have found paper to be :
"Well-motivated paper shedding light on the biomedical applications for minor languages."
Demonstrates impressive performance improvement (up to 10+ points) in cross-lingual zero-shot and few-shot scenarios.
Providing a valuable insight that knowledge (which is usually multilingual in the KB) is essential for bridging multilingual abilities."

Furthermore, the authors have been very proactive with sharing additional analysis requested by the reviewers. We suggest the authors to include the same in the revised draft, e.g.
- Performance of the model when there are fewer tokens at each granularity level
- Effectiveness of each granularity separately,

---

### Decision · Program_Chairs · 2023-10-07

**Decision:**

Accept-Findings

**Comment:**

Reviewers have found paper to be :
"Well-motivated paper shedding light on the biomedical applications for minor languages."
Demonstrates impressive performance improvement (up to 10+ points) in cross-lingual zero-shot and few-shot scenarios.
Providing a valuable insight that knowledge (which is usually multilingual in the KB) is essential for bridging multilingual abilities."

Furthermore, the authors have been very proactive with sharing additional analysis requested by the reviewers. We suggest the authors to include the same in the revised draft, e.g.
- Performance of the model when there are fewer tokens at each granularity level
- Effectiveness of each granularity separately,